# Dynamic Programming BN Structure Learning Algorithm Integrating Double Constraints under Small Sample Condition

**DOI:** 10.3390/e24101354

**Published:** 2022-09-24

**Authors:** Zhigang Lv, Yiwei Chen, Ruohai Di, Hongxi Wang, Xiaojing Sun, Chuchao He, Xiaoyan Li

**Affiliations:** 1School of Mechatronic Engineering, Xi’an Technological University, Xi’an 710021, China; 2School of Electronic Information Engineering, Xi’an Technological University, Xi’an 710021, China; 3General Office, Northwest Institute of Mechanical and Electrical Engineering, Xianyang 712099, China

**Keywords:** Bayesian network, prior knowledge, dynamic programming, edge constraint, path constraint

## Abstract

The Bayesian Network (BN) structure learning algorithm based on dynamic programming can obtain global optimal solutions. However, when the sample cannot fully contain the information of the real structure, especially when the sample size is small, the obtained structure is inaccurate. Therefore, this paper studies the planning mode and connotation of dynamic programming, restricts its process with edge and path constraints, and proposes a dynamic programming BN structure learning algorithm with double constraints under small sample conditions. The algorithm uses double constraints to limit the planning process of dynamic programming and reduces the planning space. Then, it uses double constraints to limit the selection of the optimal parent node to ensure that the optimal structure conforms to prior knowledge. Finally, the integrating prior-knowledge method and the non-integrating prior-knowledge method are simulated and compared. The simulation results verify the effectiveness of the method proposed and prove that the integrating prior knowledge can significantly improve the efficiency and accuracy of BN structure learning.

## 1. Introduction

Many problems in the real world face uncertainty factors, and artificial intelligence today deals with problems of uncertainty, such as image recognition, speech recognition, intelligent decision-making, and so on. A Bayesian Network (BN) [1], as a type of Graphical Model, has become a powerful tool to treat uncertainty problems because of its strict mathematical foundation, visual and understandable graphic topological structure, as well as a natural expression of reality problems. In recent years, Bayesian Networks have been successfully applied in various fields such as medical diagnosis [2,3], fault diagnosis [4], decision analysis [5], gene analysis [6,7], target identification [8], threat assessment [9,10], and system reliability analysis [11,12].

However, before a BN is used to solve problems in engineering practice, the BN structure needs to be constructed first. Compared with the approximate BN structure search algorithm based on constraint and a heuristic algorithm, the accurate solution of the BN structure learning has recently become a popular topic in academic research. The accurate solution includes the branch and bound method [13], integer programming [14,15], and Dynamic Programming (DP) [16,17]. Although the traditional BN structure learning algorithm based on DP can obtain the global optimal solution, the acquired structure is inaccurate when the sample does not completely contain the information of the real structure, especially when the sample size is small. The complexity problem is also the bottleneck faced by the current DP method. However, in reality, there is a lot of deterministic prior knowledge available in BN modeling. The prior-knowledge distribution of the BN structure is learned to put forward a method of Bayesian network model averaging [18]. The BN structure learning is transformed into a constrained objective function extremum problem with the node order in [19]. Campos [20] considers various deterministic constraints, analyzes the interaction between constraints, and realizes it by the hill climbing method and PC algorithm. The Nicholson [21] Incorporating expert elicited structural information in the CaMML causal discovery program. The results show that with prior knowledge, CaMML has excellent properties. Castelo [22] conducted BN structure learning by specifying the prior knowledge. Borboudakis [23] took the probability of edge and path existence as prior knowledge through rigorous mathematical derivation and conducted structured learning through the BD score and hill-climbing method. The node order prior knowledge is integrated into the process of dynamic programming in [24]. The path constraints are used to learn the BN structure with integer programming [25]. Li proposed a constraint-based hill-climbing approach to incorporate all these constraints [26]. Cussens [27] considered integer linear programming (ILP) as constrained optimization and treated all constraints as cutting planes.

As can be seen, the use of prior knowledge can not only improve the learning accuracy but also the learning efficiency. However, no one has ever studied the use of edge and path prior knowledge in the process of DP structure learning. Therefore, this paper proposes a BN structure learning algorithm based on DP, which combines expert prior knowledge and sample information effectively. The proposed algorithm incorporates edge constraints and path constraints to limit the search process of DP and delete parts of the planning space, so that all search processes meet the prior-knowledge requirements, thus reducing the complexity of the algorithm.

The rest of the paper is organized as follows. Section 2 introduces the theoretical basis of a Bayesian Network. Section 3 introduces in detail the dynamic programming BN structure learning algorithm integrating prior knowledge. In Section 4, the algorithm proposed in this paper is simulated and analyzed in terms of effectiveness and complexity. Section 5 is the conclusion.

## 2. Theoretical Basis of Bayesian Network

Prior to the general definition of Bayesian networks, several basic concepts in graph theory need to be introduced.

X and Y are two nodes in the directed graph G. X→Y means there is an edge from X to Y where X is called the parent node of Y while Y is called the child node of X. For any node X, $Ch\left(X\right)$ represents the set of all child nodes while $Pa\left(X\right)$ represents the set of all parent nodes of X. If X has no parent node, then X is a root node. The set of all root nodes of G is $Root\left(G\right)$. If X has no child node, then X is a leaf node. The set of all leaf nodes of G is $Leaf\left(G\right)$. If there are k nodes, i.e., $X_{1},\ldots ,X_{k}$ in G, and for each i=1,…,k−1, there is $X_{i}\to X_{i+1}$, then there is a directed path from $X_{i}$ to $X_{k}$, marked as $X_{1}\Rightarrow X_{k}$. For any X⇒Y, X is called the ancestor node of Y while Y is the descendant node of X. Likewise, $An\left(X\right)$ represents the set of all ancestor nodes of X and $De\left(X\right)$ represents the set of all descendant nodes. If there is a node in G, and the node is its own ancestor node, then the graph has a directed cycle. If the directed graph does not have any directed cycle, then it is a Directed Acyclic Graph (DAG).

A Bayesian Network consists of a DAG and a Conditional Probability Table (CPT), and its complete definition is as follows:

**Definition** **1** **[28].***A Bayesian Network is a binary group 1*$\langle G,\Theta \rangle$*, in which*$G=\left(V,E\right)$*represents the structure of the Bayesian network, a DAG where*$V=\left\{X_{1},X_{2},\ldots ,X_{n}\right\}$*represents a set of random variables, and*E*is a directed edge set indicating the nature of causal association between variables.*$\Theta =\left\{P(X_{i}|Pa(X_{i})):X_{i}\in V\right\}$*is a Conditional Probability Table (CPT)*.

**Definition** **2** **[28].**
*(node order) A node order*

o

*refers to the linear arrangement of some variables in which*

$X_{i}≺X_{j}$

*means*

$X_{i}$

*is in front of*

$X_{j}$

*. The node order*

o

*is the node order of*

G

*. If and only if for arbitrary*

$\left\{X_{i},X_{j}\right\}\subset vari\left(o\right)$

*there is*

$X_{i}≺X_{j}$

*in*

o

*, then*

$X_{j}$

*cannot be an ancestor node of*

$X_{i}.$



**Theorem** **1** **[28].***With BIC score as the criterion, in an optimal Bayesian network, any node has at most*$\left\lfloor \log\left(2N/\log N\right)\right\rfloor$*parent nodes, where*N*is the number of samples. This article refers to*$\left\lfloor \log\left(2N/\log N\right)\right\rfloor$*as*$n_{mp}$*(max number parents)*.

## 3. BN Structure Learning Algorithm for Dynamic Programming Integrating Prior-Knowledge

### 3.1. Dynamic Programming Algorithm

A Bayesian network structure learning algorithm based on dynamic programming is a process of accurately solving mathematical programming problems, and with exponential computational complexity, it is limited by the number of nodes. The state transition equations of dynamic programming are:(1)maxScoreV=maxX∈VmaxScoreV\X+maxScoreX,V\X,
(2)bestscoreX,V\X=maxScoreX,V\X=maxPaX∈V\XScoreX,PaX,
where V is a set of variables, X is a leaf node in the optimal structure, and Score• is a decomposable scoring function [29]. Equations (1) and (2) connect the relationship between the whole structure and its substructures, and the optimal network on the remaining nodes V\X is recursively constructed through the above process until the remaining nodes are a variable. All the child node sets form a Hasse Diagram, showing the whole process of dynamic programming. When the DP algorithm calculates from top to bottom, the root node is determined first, and then the leaf nodes that are gradually added to the remaining nodes are universal set variables. When the DP algorithm calculates from bottom to top, leaf nodes are determined first, and then the root nodes that are gradually added to the remaining nodes are empty set variables. Because a Hasse Diagram contains the node order information of the network, it is also called the Order Graph. There is another similar graph called the parents graph [17]. Figure 1 shows a node order graph with the number of nodes as *n* = 4 and the parent node graph of node X1.

### 3.2. Expression of Constraints

In this paper, deterministic prior knowledge is directly transformed into some constraints. Prior knowledge and prior constraints are equivalent concepts. For convenience of expression, constraints are used to refer to prior knowledge later. C is used in this article to refer to a set of constraints representing edges or paths, which are expressed as follows:
X→Y means X is the parent node of Y. X→Y means X cannot be the parent node of Y. $edge\left(X,Y\right)$ is used to express any edge constraint between X and Y;X⇒Y means X is the ancestor node of Y. X⇒Y means X cannot be an ancestor node of Y. In these two cases, Y is called a tail node and is a head node. $path\left(X,Y\right)$ is used to express any X path constraint between X and Y;Suppose there is an arbitrary node order o and constraint set C in which o and C are consistent. If and only if for arbitrary $\left\{X_{1},X_{2}\right\}\subset \left(vari\left(o\right)\cap vari\left(C\right)\right)$, there is $X_{1}≺X_{2}$ in o, then $X_{2}$ cannot be an ancestor node of $X_{1}$ in C.


This paper uses constraints in the following two steps: (1) to limit the construction process of the node order graph. Specifically, some illegal nodes are deleted from the node order graph, which can reduce the complexity, especially the space complexity. (2) A sparse parent node graph and query algorithm are constructed, so that the results of the optimal parent node query can satisfy the constraints. Theorem 2 is given as the basis for realizing constraints.

**Theorem** **2.***In a given set of constraints on edges or paths*$edge\left(X_{1},Y_{1}\right)\in C$*,*$path\left(X_{2},Y_{2}\right)\in C$*, for any optimal substructure*G*in the dynamic programming process, if*$\left\{X_{1},Y_{1}\right\}\subset vari\left(G\right)$*, then there must be*$edge\left(X_{1},Y_{1}\right)\in G$*. Similarly, if*$\left\{X_{2},Y_{2}\right\}\subset vari\left(G\right)$*, then there must be*$path\left(X_{2},Y_{2}\right)\in G$.

**Proof** **of** **Theorem** **2.**If $edge\left(X_{1},Y_{1}\right)\in C$ and $\left\{X_{1},Y_{1}\right\}\subset vari\left(G\right)$, and if there is no $edge\left(X_{1},Y_{1}\right)$ in G, then due to the non-aftereffect property of the dynamic programming method, all the extended structures G will not satisfy constraints C, so there must be $edge\left(X_{1},Y_{1}\right)$ in G. $path\left(X_{2},Y_{2}\right)\in C$ can also be proven in the same way. The proof is completed. □

### 3.3. Integrating Constraints of Edge

#### 3.3.1. Pruning Node Order Graph

With the given constraints of edges $X_{1}\to X_{2}$, node $\left\{X_{2}\right\}$ in the node order graph needs to be deleted because it violates the constraint: the structures produced by the optimal substructure of this node all satisfy the node order $X_{2}≺X_{1}$ which is obviously inconsistent with constraints $X_{1}≺X_{2}$, so it is unnecessary to calculate node $X_{2}$ when constructing the node order graph. As can be seen from the above example, if a node in the node order graph violates a constraint, it needs to be deleted. The theorem is given as follows:

**Theorem** **3.***In a given constraint set*C*, there is a node*U*and its set of node order*$o_{U}$*in the node order graph, then*U*needs to be deleted from the node order graph if and only if there is such a*$\left\{X_{1},Y_{1}\right\}\subset vari\left(C\right)$*, satisfying the condition that*$X_{1}\Rightarrow X_{2}$*can be inferred from*C*and there are*$X_{2}\in U$*,*$X_{1}\notin U$.

**Proof** **of** **Theorem** **3.**In subsequent nodes of any, U, $X_{1}$ is added as a leaf node, so obviously for any $o\in o_{U}$, there is $X_{2}≺X_{1}$, which is inconsistent with $X_{1}\Rightarrow X_{2}$. Suppose for any $X_{1}\Rightarrow X_{2}$ relationship in C, there is no $X_{2}\in U$, $X_{1}\notin U$, then $o_{1}$ in $vari\left(o_{1}\right)=U$ is consistent with C, and $o_{2}$ in $vari\left(o_{2}\right)=V\backslash U$ is consistent with C. Moreover, for any $X_{1}\Rightarrow X_{2}$, there is no $X_{2}\in vari\left(o_{1}\right)$ or $X_{1}\in vari\left(o_{2}\right)$. So o made up of $o_{1}$ and $o_{2}$ is consistent with C. The proof is completed. □

**Theorem** **4.***In a given edge constraint set*C*and the variable set*V*of the problem domain, when traversing to any node*U*during the construction of node order graph, make*$G_{s}=sub\left(G_{C},vari\left(C\right)\backslash U\right)$*. If the resulting new node*U∪X*of*U*must satisfy*$X\in \left(V\backslash vari\left(C\right)\cup root\left(G_{s}\right)\right)$*, then all the constructed nodes in the last node order graph satisfy the constraint*C*and all deleted nodes violate the constraint*C.

**Proof** **of** **Theorem** **4.**The node order graph is constructed from an empty set, which satisfies the constraint C. At this time, no node in the node order graph is deleted. When traversing to any node U, suppose all the existing nodes in the node order graph satisfy C and all the deleted nodes violate C. Then, it is necessary to prove that the new nodes constructed in the node order graph satisfy the constraints and the deleted nodes violate the constraints.

The newly constructed nodes satisfy the constraints: any constructed new node is U∪X, in which $X\in \left(V\backslash vari\left(C\right)\cup root\left(G_{s}\right)\right)$ and $G_{s}=sub\left(G_{C},vari\left(C\right)\backslash U\right)$. If $X\in \left(V\backslash vari\left(C\right)\right)X_{1},\ldots ,X_{n}$, because U satisfies the constraint, and the new variable X has nothing to do with constraint C, then obviously U∪X also satisfies the constraint. If $X\in \left(root\left(sub\left(G_{C},vari\left(C\right)\backslash U\right)\right)\right)$, the newly added variable X is the remaining variable in $vari\left(C\right)$ and it is the root node in the subgraph of the remaining variables of the edge constraint graph. So there is no such ancestor node $Y\in vari\left(C\right)\backslash \left(U\cup X\right)$ of X in the remaining nodes, causing there to be Y⇒X in C. Therefore, according to Theorem 3, any newly added node satisfies the constraints.

No deleted node satisfies the constraints. Suppose H is deleted: H can be remade by combining H\X with X, with X as any variable in H. If there is H\X and H\X is the node that satisfies the constraint, then X is a non-root node in the corresponding subgraph, but in this subgraph, there must be a corresponding root node Y. There is Y⇒X and Y∉H, so according to Theorem 3, H does not satisfy the constraint. When there is no arbitrary X to make H\X satisfy the constraint and if H satisfies the constraint, according to Theorem 3, there is a Y in H and X⇒Y. Then, a node order of constraint C, i.e., X≺Y≺…≺Z, can be constructed by using H∩variC. Take the last variable Z: if Z does not exist, then Z=Y and H\Z satisfy the constraint, which contradicts the condition. Therefore, the hypothesis is invalid, which proves that the arbitrarily deleted node does not satisfy the constraints. The proof is completed. □

Theorem 3 provides the basis for pruning the node order graph. The most direct way for the pruning is to make judgments on each node U so that they satisfy Theorem 3. However, even the simplified judgment algorithms cannot perform with the best efficiency. Therefore, Theorem 4 proposes a method to construct a node order graph, so that all nodes in the graph satisfy the constraint. According to the method in Theorem 4, we can make full use of the constraint to prune the node order graph, reduce the space complexity, and obtain the optimal structure after the node order graph is constructed.

As the score of sets in the node order graph needs to be queried repeatedly, in order to increase the efficiency of sets querying, this paper designs the hash function of sets in which different sets correspond to different hash values. The hash function is designed as follows: Suppose the set of all variable in the problem domain is $\left\{X_{1},\ldots ,X_{n}\right\}$. Set binary number b with the number of digits as n. For a set U in the node order graph, if $X_{i}\in U$, then set the *i*th place of b to 1, otherwise set it to 0. Finally, convert b to decimal which will be the corresponding hash value.

The specific algorithm flow of the node order graph construction is shown in Algorithm 1.
**Algorithm 1.** Construction algorithm of node order graph.**Constructing Node Order Graph Based on Edge Constraint****Input:**SPG -Sparse parent node graph, GC -edge constraint graph**Output:**
G-Global optimal structure1. PreviousLayer.HashTable←∅, PrevioUsLayer[∅].score←0, VC=variC2. forLayer←1 to n do3.     **for** each node U In the PreviousLyaer **do**
4.         Vr←Vc\U5.         Gr← Removing variables of Vr And their relative arcs in Gc6.         R← Root variables of Gr7.             **for** each variable X∈V\U∪Vr∪R **do**
8.              [bestparents bestscore]←GetBestScoreX,U,C,SPG9.            curscore←U.score+bestscore10.               **if**
NewLayer[U∪X] Is null 11.                NewLayer[U∪X]←curscore,parents,HashTable12.             **else if**
curscore > NewLayer[U∪X].score then 13.             NewLayer[U∪X]←curscore,parents,HashTable14.             **end if**15.          **end for**16.     **end for**17.    PreviousLayer←NewLayer18.   **end for**19.  G←querying PreviousLayer.HashTable

#### 3.3.2. Construction and Query of Sparse Parent Node Graph

The construction algorithm of the sparse parent node graph is as follows: As the sparse parent node graph stores the information of the first $n_{mp}$ layers of nodes in the complete parent node graph, we first construct a complete parent node graph based on the constraint according to Theorem 4, and then store it in the sparse parent node graph every time a node is constructed. When the first $n_{mp}$ layers are completely constructed, the sparse parent node graph will be obtained. Here is an example to illustrate how to construct a complete parent node graph based on this constraint.

Figure 2 gives some constraints of variables. To calculate a node of $X_{3}$ in the graph, such as the node $\left\{X_{1},X_{2}\right\}$, we only need to compare the scores of $\left\{X_{1}\right\}$ and $\left\{X_{2}\right\}$, i.e.,$score_{\left\{X_{1}\right\}}$ $score_{\left\{X_{2}\right\}}$ and $score\left(X_{3},\left\{X_{1},X_{2}\right\}\right)$ when there is no constraint. Moreover, because $X_{1}$ is the parent node in the constraint, only $score_{\left\{X_{1}\right\}}$,$score\left(X_{3},\left\{X_{1},X_{2}\right\}\right)$ are compared now. In addition, the crossed nodes in Figure 2b do not need to be solved because these nodes contain variables $X_{4}$ and $X_{4}$ is by no means the parent node of $X_{3}$.

Based on the above ideas, the specific algorithm flow of the sparse parent node graph construction, defined as **PBDP****-EDGE,** is shown in Algorithm 2.
**Algorithm****2.** Construction algorithm of sparse parent node graph.**Constructing Sparse Parent Node Graph Based on Edge Constraint****Input:** V-set of all variables,C-set of constraints,score(.,.)-decomposable score function value**Output:** SPG-Sparse parent node graph1.  **for**
X∈V **do**2.    scoreX,parentsX←∅3.      **for**
layer←0 to n **do**
4.          **for** each node U Such that U∈V\X&U==layer **do**
5.             BestScoreX,U=maxY∈P,Y∉PaXBestScoreX,U\Y6.               **if** U∩noPaX==∅&&scoreX,U>BestScoreX,U7.            BestScoreX,U←scoreX,U8.            Append [*score_X_, parents_X_*] with BestScoreX,U,bitnarizeU 9.               **end if**10.        **end for**11.    **end for**12.    Sort scoreX,parentsX with scoreX In descending 13. **end for**14. **return**
*SPG* ← [*score*., *patrnts*.]

The query algorithm idea of the sparse parent node graph is as follows: Suppose δ is a query constraint of X, that is, all the possible $pa\left(X\right)$ in U must satisfy the constraint δ. In other words, the front set of $parents_{X}$ that satisfy δ is the best parent node set in U. Furthermore, query constraint δ must satisfy the following two conditions: (1) Y⊂U, (2) $CPa\left(X\right)\cap U\in Y$, $Y\cap CNPa\left(X\right)=\emptyset$, in which $CPa\left(X\right)$ means it is the set of parent nodes of X and $CNPa\left(X\right)$ means it is not.

The specific implementation is as follows: First, set a bit array validX of all 1s and with the same length as parentsX. Then, according to the first condition in δ, first do validX&∼parentsXXi for each Xi that satisfies Xi∈V\U. Then, according to the second condition in δ, do validX&∼parentsXXi for each Xi that satisfies Xi∈CPaX∩U. The purpose of this step is to ensure that all the remaining sets include all the variables in CPaX∩U. Finally, conduct validX&∼parentsXXi for each Xi that satisfies Xi∈CNPaX∩U. This step eliminates the sets which include variables in any CNPaX∩U, and the front set in the remaining sets is the best parent node set. Algorithm 3 shows the specific algorithm of the best parent node set query.
**Algorithm 3.** Query algorithm of best parent node set.**The Optimal Parent Node Set Based on Query Constraints****Input:** V-set of all variables, C-set of constraints, SPG-sparse parent node graph**Output:** bestsparents.,.-The best parent node set, bestscore.,.-The corresponding score1. valid←allScoresX2. **for** each Y1∈PaX∩U **do**
3.     valid←valid&parentsXY14. **end for**5. **for** each Y2∈V\U∪noPaX\PaX **do**
6.     valid←valid&~parentsXY27. **end for**8. index←firstSetBitvalid9. **return**
scroreXindex, parentsXindex

Here is an example to illustrate the implementation process. As shown in Table 1, from $\left\{X_{1},X_{2},X_{4},X_{5}\right\}$, find the best parent node sets of $X_{3}$, $CPa\left(X_{3}\right)=\left\{X_{1},X_{2}\right\}$ and $CNPa\left(X_{3}\right)=\left\{X_{4}\right\}$. At this point, the remaining candidate sets in the table all satisfy the first condition of δ, that is, they all are subsets of $\left\{X_{1},X_{2},X_{4},X_{5}\right\}$. If there is no constraint, $\left\{X_{1},X_{5}\right\}$ will be the best parent node set. Next, we need to realize the second condition. Because $\left\{X_{1},X_{2}\right\}=CPa\left(X\right)\cap U$, do $parents_{X}^{X_{1}}\&parents_{X}^{X_{2}}$. The result is shown in the seventh row of the table, in which a value of 1 means that all sets contain $\left\{X_{1},X_{2}\right\}$. Because $CNPa\left(X_{3}\right)=\left\{X_{4}\right\}$, find $∼parents_{X}^{X_{4}}$. The result is shown in the eighth row of the table in which the value of 1 means none of them contains $X_{4}$. Finally, sum the seventh and the eighth row to obtain the final $valid_{X}$. The first set $\left\{X_{1},X_{2},X_{5}\right\}$ is the best parent node set satisfying the constraint.

### 3.4. Integrating Path Constraints

#### 3.4.1. Pruning Node Order Graph

The algorithm of the pruning node order graph by path constraint is the same as that by edge constraint. First, construct a constraint graph $G_{C}$, then use the algorithm in Table 1 to prune the node order graph, wherein path constraint graph $G_{C}$ of the constraint set C is a directed acyclic graph containing variable $vari\left(C\right)$, and for arbitrary $\left\{X_{1},X_{2}\right\}\subset vari\left(C\right)$, there is an edge $X_{1}\to X_{2}$ between $X_{1},X_{2}$ in $G_{C}$, if and only if $\left(X_{1}\Rightarrow X_{2}\right)\in C$.

#### 3.4.2. Construction and Query of Sparse Parent Node Graph

As the parent node of Y must contain at least one X or one descendant node of X, when constructing the sparse parent node graph, for Y, it is necessary to store all the parent node sets with the number of variables below $n_{mp}$. For other variables, the sparse parent node graph is constructed according to the unconstrained condition. The specific construction algorithm of the sparse parent node graph, defined as **PBDP****-PATH,** is shown in Algorithm 4.
**Algorithm 4.** Construction algorithm of sparse parent node graph.**Constructing Sparse Parent Node Graph Based on Path Constraints****Input:**V-set of all variables,C-set of constraints,score(.,.)-decomposable score function value**Output:**SPG-sparse parent node graph1.**for**
X∈V **do**2.      **if**
$X\in end\left(C\right)$ **do**
3.         Construct Full Sparse Parent Graph (V, X, score(.,.)) 4.      **else do**5.        
Construct Sparse Parent Graph without Constraints (V,X, score(.,.))6..      **end if**7. **end for**8. **return** SPG←score·,parents·9. **Function** Construct Sparse Parent Graph without Constraints (V, X, score(.,.))10.      scoreX,parentsX←∅
11.       **for**
layer←0 to n **do**
12.          
**for** each node P such that $P\in V\backslash \left\{X\right\}\&\left|P\right|==layer$ **do**
13.          
BestScoreX,P=maxY∈PBestScoreX,P\Y14.           **if** scoreX,P>BestScoreX,P
15.             BestScoreX,P←scoreX,P
16.             append scoreX,parentsX with ScoreX,P,bitnarizeP
17.         
**end if**18.        
**end for**19.      **end for**20.      sort with scoreX in descending21.       **return** scoreX,parentsX22. **end function**23. **Function** Construct Full Sparse Parent Graph (V, X, score(.,.))24.       scoreX,parentsX←∅25.       **for** each P∈V\X **do**
26.          
Append scoreX,parentsX with ScoreX,P,bitnarizeP27.       **end for**28.       sort with scoreX in descending29.       **return**
scoreX,parentsX30. **end function**

The query algorithm idea of path constraint is as follows: For a given path constraint X⇒Y, to find the best parent node set S of Y in U, if X∈U, there is at least one $Z\in \left\{X\cup des\left(X\right)\right\}$ to make Z∈S. $des\left(X\right)$ is the descendant node of X in the structure of U. If X⇒Y, then there is Z∉S for all $Z\in \left\{X\cup des\left(X\right)\right\}$.

The specific query method is as follows: Initialize a bit array validX of all 1s and with the same length as parentsX. Conduct validX&∼parentsXXk for each Xk that satisfies Xk∈V\U. For each Xi, set an auxiliary bit array Cvalid of all ones and find the descendant node desXi of Xi. For each Z∈Xi∪desXi, perform the OR operation Cvalid|parentsXZ. Finally, perform the AND operation valid←valid&Cvalid. For each Xj, find desXj. For each Z∈Xi∪desXi, perform the AND operation valid←valid & parentsXZ. Algorithm 5 shows the specific algorithm flow of the best parent node set query.
**Algorithm 5.** Query algorithm of best parent node set.**The Best Parent Node Set Based on the Path Constraint Query****Input:** V-set of all variables, C set of path constraints, SPG-sparse parent node graph**Output:** bestsparents.,.-the best parent node set, bestsCore.,.-the corresponding score1.valid←allScoresX2. **for** each **do**
3.     valid←valid&~parentsXYi4. **end for**5. **for** each Yj such that Yj⇒X∈C **do**6.         Cvalid←allScoresX7.         **for** each S Holding that Yj⇒S in G **do**
8.           Cvalid←Cvalid|parentsXS9.           Cvalid←Cvalid|parentsXYi
10.     **end for**11.     valid←valid&Cvalid12. **end for**13. **for** each Yk such that Yk⇒X∈C **do**14.        
**for** each S Holding that Yk⇒S in G **do**
15.           
valid←valid&~parentsXS16.         **end for**17. **end for**18. index←firstSetBitvalid19. **return** scroresXindex

An example is given below to illustrate the implementation process.

Figure 3 is an example of path constraints, in which C is $X_{1}\Rightarrow Y$ and $X_{2}\Rightarrow Y$. At this point, we need to find the best parent node set S of Y from $U=\left\{X_{1},X_{2},X_{3},X_{4}\right\}$. Table 2 shows the specific solution process. In the table, parent node sets are selected with part of them as the subset of U, so the first condition of δ has been satisfied. At this point, if there is no constraint, $\left\{X_{2},X_{4}\right\}$ will be the best parent node set. When a constraint is given, perform the OR operation for the line where the elements of $\left\{desX_{1}\cup X_{1}\right\}$ are and obtain $Cvalid_{1}$, as in line 7. Perform OR operation for the line where the elements of $\left\{desX_{2}\cup X_{2}\right\}$ are and obtain $Cvalid_{2}$, as in line 8. Then, perform the AND operation $valid\leftarrow valid\&Cvalid_{1}\&Cvalid_{2}$. At this time, valid equals 1, which shows that there are elements both from $\left\{desX_{1}\cup X_{1}\right\}$ and $\left\{desX_{2}\cup X_{2}\right\}$; therefore $\left\{X_{3},X_{4}\right\}$ is the best solution at this time.

## 4. Algorithm Simulation and Analysis

### 4.1. Validity Verification

In this section, in order to verify the effectiveness, first, an 18-node network is generated by using Matlab constructor. Then, the constructed network, Asia network, and Sachs network are simulated and verified with 20 samples. In order to verify that this method can really integrate constraints, some extreme simulation conditions are set.
1.The simulation is carried out with the Asia network. All the edge prior knowledge is given, which is verified by the PBDP-EDGE structure. Part of the path prior knowledge is given, specifically 1⇒6, 2⇒6, 2⇒8, 3⇒7, 3⇒8, 4⇒7, and 4⇒8, which is verified by the PBDP-PATH structure. The results are shown in Figure 4.

The real network structure of the Asia network is shown in Figure 4a. It can be seen from Figure 4b that training samples contain very little information and can only learn a few edges, and a complete structure cannot be constructed. It can be seen from Figure 4c that the correct structure can be learned even if the sample size is small, which demonstrates the correctness and effectiveness of the integrating edge prior-knowledge algorithm proposed in this paper. It can be seen from Figure 4d that it is obvious that all the learned structures contain these paths (1⇒6, 2⇒6, 2⇒8, 3⇒7, 3⇒8, 4⇒7, and 4⇒8).
2.The simulation is carried out with the Sachs network. Part of the edge prior knowledge is given, which is verified by the PBDP-EDGE structure. Part of the path prior knowledge is given, specifically 1⇒2, 1⇒4, 1⇒5, 1⇒7, 1⇒8, 2⇒5, 2⇒8, 3⇒4, 4⇒6, 5⇒6, and 9⇒11, which is verified by the PBDP-PATH structure. The results are shown in Figure 5.

The real network structure of the Sachs network is shown in Figure 5a. As can be seen from Figure 5b, the training sample contains little information, only a few edges can be learned, and a complete structure cannot be constructed. It can be seen from Figure 5c that partial correct structures can be learned even if the sample size is small, indicating the correctness and effectiveness of integrating the edge prior-knowledge algorithm proposed in this paper. It can be seen from Figure 5d that it is obvious that all the learned structures contain these paths (1⇒2, 1⇒4, 1⇒5, 1⇒7, 1⇒8, 2⇒5, 2⇒8, 3⇒4, 4⇒6, 5⇒6, and 9⇒11).
3.The simulation is carried out with the Constructed network. Part of the edge prior knowledge is given, which is verified by the PBDP-EDGE structure. Part of the path prior knowledge is given, specifically 1⇒4, 1⇒17, 2⇒18, 2⇒13, 2⇒5, 3⇒5, 3⇒9, 6⇒8, 5⇒10, 5⇒12, 7⇒5, 10⇒13, 13⇒9, and 15⇒10, which is verified by the PBDP-PATH structure. The results are shown in Figure 6.

The real structure of the Constructed network is shown in Figure 6a. As can be seen from Figure 6b, the training samples contain little information, only a few edges can be learned, and a complete structure cannot be constructed. It can be seen from Figure 6c that partial correct structures can be learned even if the sample size is small, indicating the correctness and effectiveness of the integrating constraints of the edge algorithm proposed in this paper. It can be seen from Figure 6d that it is obvious that all the learned structures contain these paths (1⇒4, 1⇒17, 2⇒18, 2⇒13, 2⇒5, 3⇒5, 3⇒9, 6⇒8, 5⇒10, 5⇒12, 7⇒5, 10⇒13, 13⇒9, and 15⇒10).

Therefore, the above simulation results can prove that the method proposed in this paper is correct and reliable and can be realized no matter what kind of prior knowledge is given.

### 4.2. Complexity Verification

The integrating edge constraint is simulated by the Halifinder network, a large-scaled network, and half of the real edges are randomly selected as prior knowledge. The training sample size is 200, 500, and 1000, respectively. Table 3 shows the simulation results. PBDP (Priors Based DP) indicates the integrating prior-knowledge method, which is measured in seconds. The space cost refers to the size of the array to be set, and the proportion represents the time and space ratio between the PBDP method and DP method.
The path constraint is simulated in the same way, with the results shown in Table 4.

It can be seen from Table 3 and Table 4 that the integrating edge constraint and path constraint can not only improve the scores, but also effectively reduce the complexity of time and space. To sum up, this method can use edge constraints and path constraints to effectively reduce the time and space complexity of the Dynamic Programming algorithm and improve its timeliness significantly.

## 5. Conclusions

In this paper, the specific process of dynamic planning is analyzed, and its restrictive relationship with edge constraints and path constraints is determined. The prior constraints are used to restrict and guide each link in dynamic planning, and deterministic prior knowledge is integrated into the dynamic planning of BN structure learning. The BN structure learning algorithm of dynamic planning integrating prior knowledge is proposed, and the specific implementation is described in detail. Simulation results show that this algorithm can use edge prior knowledge and path prior knowledge to effectively reduce the time and space complexity of the dynamic programming algorithm. It also reveals the complementary relationship between prior knowledge and learning in BN modeling, that is, only by making full use of prior knowledge and training sample information can an ideal model be obtained. This paper also provides some implications for the breaking through of the node number in the dynamic programming method.

## Figures and Tables

**Figure 1 entropy-24-01354-f001:**
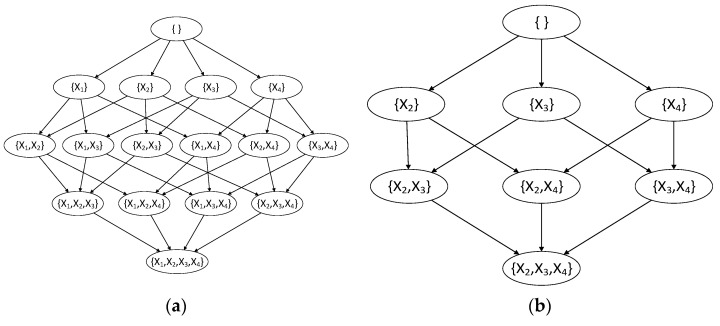
Node order graph and parent node graph of *X*_1_: (**a**) Node order graph of four nodes; (**b**) Parent node graph of *X*_1_.

**Figure 2 entropy-24-01354-f002:**
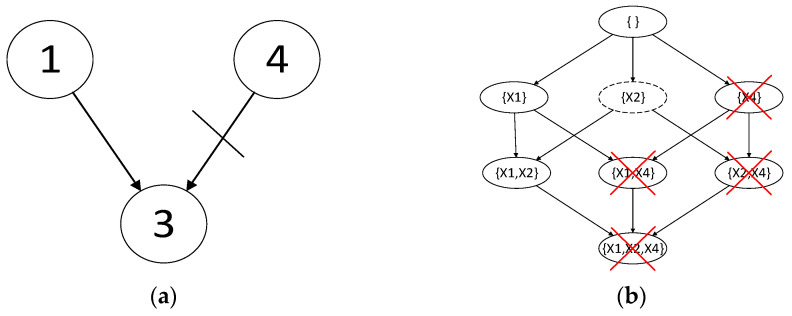
Construction of complete parent node graph of $X_{3}$ with given edge constraint: (**a**) Edge constraint; (**b**) Construction of complete parent node graph of $X_{3}$.

**Figure 3 entropy-24-01354-f003:**
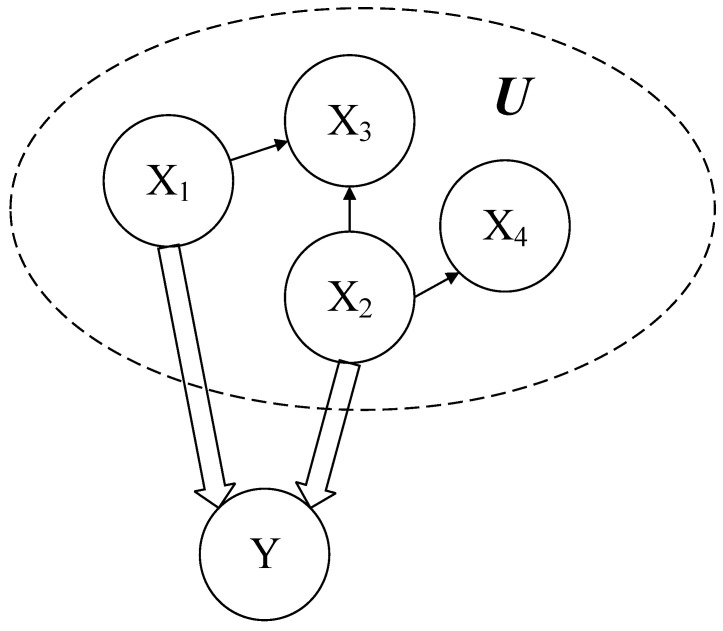
Example of path constraints.

**Figure 4 entropy-24-01354-f004:**
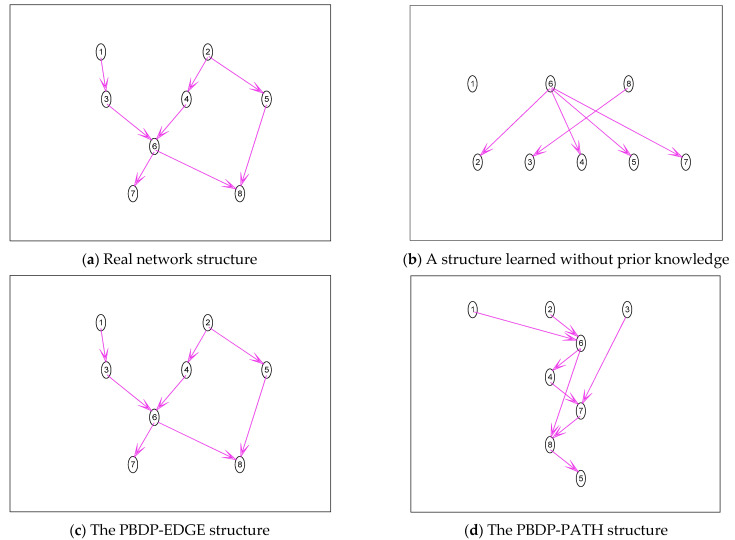
Asia network structure simulation diagram.

**Figure 5 entropy-24-01354-f005:**
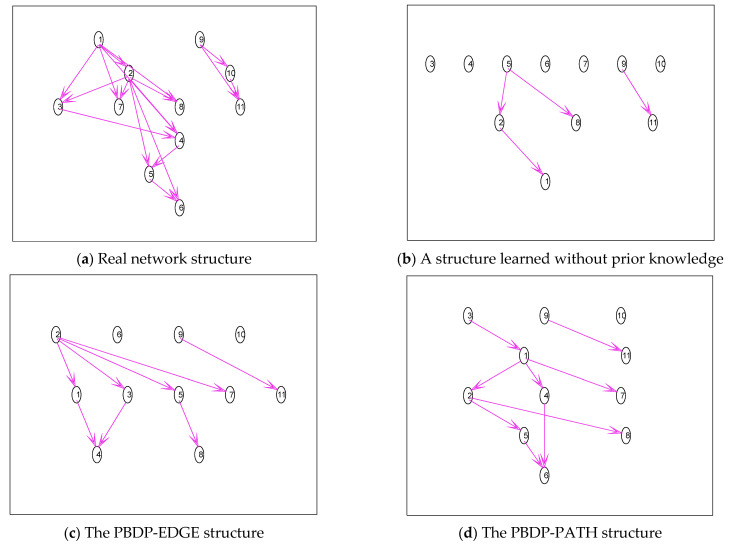
Sachs network structure simulation diagram.

**Figure 6 entropy-24-01354-f006:**
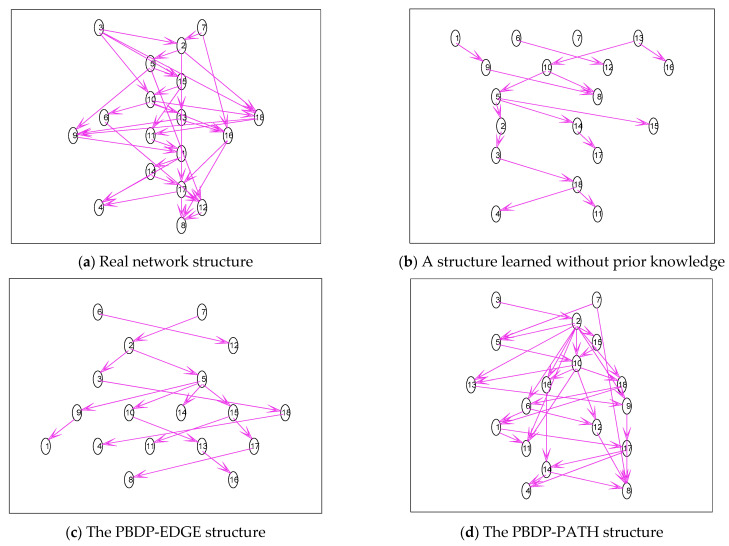
Constructed network structure simulation diagram.

**Table 1 entropy-24-01354-t001:** Example of query process based on constraints.

1	$parents_{X_{3}}$	$\left\{X_{1},X_{5}\right\}$	$\left\{X_{1},X_{2},X_{4}\right\}$	$\left\{X_{1},X_{2},X_{5}\right\}$	{ }	$\left\{X_{1}\right\}$
2	$scores_{X_{3}}$	5	4	3	2	1
3	$parents_{X_{3}}^{X_{1}}$	1	1	1	0	1
4	$parents_{X_{3}}^{X_{2}}$	0	1	1	0	0
5	$parents_{X_{3}}^{X_{4}}$	0	1	0	0	0
6	$parents_{X_{3}}^{X_{5}}$	1	0	1	0	0
7	Operation of $CPa\left(X_{3}\right)$	0	1	1	0	0
8	$\mathrm{Operation}\;\mathrm{of}\;CNPa\left(X_{3}\right)$	1	0	1	1	1
9	valid	0	0	1	0	0

**Table 2 entropy-24-01354-t002:** Example of query process based on path constraint.

1	$parents_{Y}$	$\left\{X_{2},X_{4}\right\}$	$\left\{X_{4}\right\}$	$\left\{X_{3},X_{4}\right\}$	{ }	$\left\{X_{3}\right\}$
2	$scores_{Y}$	5	4	3	2	1
3	$parents_{Y}^{X_{1}}$	0	0	0	0	0
4	$parents_{Y}^{X_{2}}$	1	0	0	0	0
5	$parents_{Y}^{X_{3}}$	0	0	1	0	1
6	$parents_{Y}^{X_{4}}$	1	1	1	0	0
7	Cvalid $\;\mathrm{of}\;\left\{desX_{1}\cup X_{1}\right\}$	0	0	1	0	1
8	Cvalid $\;\mathrm{of}\;\left\{desX_{2}\cup X_{2}\right\}$	1	1	1	0	1
9	valid	0	0	1	0	1

**Table 3 entropy-24-01354-t003:** Simulation comparison of integrating constraints of edge.

Sample Size	Approach	PBDP-EDGE	DP	Proportion
200	PIC Score	−670,352.217	−696,271.251	
	Runtime	3723.053	31,114.242	0.12
	Space	52,223	262143	0.199
500	PIC Score	−629,520.727	−635,543.295	
	Runtime	3141.870	31,925.566	0.098
	Space	52,223	262143	0.199
1000	PIC Score	−629,520.727	−630,672.929	
	Runtime	3218.295	30,909.447	0.104
	Space	52,223	262,143	0.199

**Table 4 entropy-24-01354-t004:** Simulation comparison of integrating path constraints.

Sample Size	Approach	PBDP-PATH	DP	Proportion
200	PIC Score	−654,151.15	−684,005.61	
	Runtime	5486.113	29,359.588	0.187
	Space	44,159	262,143	0.168
500	PIC Score	−633,710.86	−637,161.52	
	Runtime	5035.867	29,785.541	0.169
	Space	44,159	262,143	0.168
1000	PIC Score	−629,561.51	−630,698.96	
	Runtime	5323.846	27,892.218	0.191
	Space	44,159	262,143	0.168

## Data Availability

Data Availability Statement: The true networks of all data sets are known, and they are publicly available (http://www.bnlearn.com/bnrepository (accessed on 15 June 2021)).

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
