# Peer review of "Dynamic Programming BN Structure Learning Algorithm Integrating Double Constraints under Small Sample Condition"

_entropy, 2022, doi:10.3390/e24101354_

Round 1

Reviewer 1 Report

Some terminology must be carefully chosen. For example, line 94 states: "Bayesian Network consists of a DAG and a parameter table". Later on this is called Conditional probability table (CPT). This is a big difference. The correct answer is CPT, not a "parameter table".

Table 4 is not explained well. Conclusion is minimal and unclear.

English requires corrections. For example,

Lines 31-32: "Many problems in the real world face uncertainty factors, and artificial intelligence  today deal [MUST BE "deals"] with problems of uncertainty...".

Line 36, this phrase does not make sense: "as natural expression of reality problems".

The list goes on.

Author Response

Responses Letter

Thank you, all. we really appreciate the insightful reviewers’ comments, given from a writing perspective. The comments push us to revisit the proposed algorithm and we made effort to address all the concerns. The manuscript has been revised accordingly and the revisions are marked in red. Detailed responses are stated as below.

---- Reviewer 1 ---

Some terminology must be carefully chosen. For example, line 94 states: "Bayesian Network consists of a DAG and a parameter table". Later on this is called Conditional probability table (CPT). This is a big difference. The correct answer is CPT, not a "parameter table".

Response: Okay, we have changed the "parameter table" to "Conditional probability table (CPT)" and marked it in red in the revised manuscript.

Table 4 is not explained well. Conclusion is minimal and unclear.

Response: The seventh row in Table 4 should be Operation of , which explains why and in detail as follows.

This table finds the optimal parent node set of , the parent node set of  is known to be , and the edge constraints are  and . The 1st row of Table 4 is the candidate parent node set of , and the 3rd to 6th rows show that these sets are all subsets of , that is, these sets satisfy the first constraint of , which can be obtained from the scores of the 2nd row. It can be seen that these candidate sets are arranged in descending order of score, and the highest score is the optimal parent node set of , so if there is no constraint,  is the optimal parent node set. Next, the edge constraint is added to the candidate set. Line 7 is to add constraint , that is, the candidate parent node set of  must contain , and line 8 is to add constraint , that is, the candidate parent node set of  must not contain . From the second constraint of , we know that the optimal parent node set must satisfy both the 7th and 8th rows, and sum the 7th and 8th rows to get the final . In , it can be seen that the candidate set that satisfies the constraints at the same time is only , that is, the set is the optimal parent node set of .

English requires corrections. For example,

Lines 31-32: "Many problems in the real world face uncertainty factors, and artificial intelligence today deal [MUST BE "deals"] with problems of uncertainty...".

Line 36, this phrase does not make sense: "as natural expression of reality problems".

The list goes on.

Response: Okay, we have checked and polished the whole paper.

Reviewer 2 Report

I read a paper about BNs and I was expecting something more difficult for the proposed algorithm to be able to handle. Specifically, if some constraints are already given, why should someone choose this method over the MMHC (Tsamardinos et al., 2006) for instance, or any other hybrid algorithm that is also implemented in an R package or Python?

To show that your algorithm recovers the true DAG with a few variables is not that big of success, especially nowadays.

The simulation studies are not at all convincing. I would like to see more cases and comparison with other algorithms. Compute some performance metrics and show that you do better, in higher dimensional settings, e.g. 50, 100, 200 variables.

Author Response

Responses Letter

Thank you, all. we really appreciate the insightful reviewers’ comments, given from a writing perspective. The comments push us to revisit the proposed algorithm and we made effort to address all the concerns. Detailed responses are stated as below.

---- Reviewer 2 ---

I read a paper about BNs and I was expecting something more difficult for the proposed algorithm to be able to handle. Specifically, if some constraints are already given, why should someone choose this method over the MMHC (Tsamardinos et al., 2006) for instance, or any other hybrid algorithm that is also implemented in an R package or Python?

Response: The MMHC algorithm found the local optimal structure in the Bayesian network structure learning. However, it didn’t find the global optimal structure. The DP algorithm selected in this paper can find the global optimal structure, but when the sample cannot completely contain the information of the real structure, especially when the sample size is small, the accurate structure cannot be obtained. Therefore, adding constraints to the algorithm can make the algorithm obtain a more accurate structure when the sample size is small. And the efficiency and accuracy of BN structure learning can also be significantly improved after adding constraints.

To show that your algorithm recovers the true DAG with a few variables is not that big of success, especially nowadays.

Response: The algorithm used in this paper is the DP algorithm, which can obtain the global optimal solution. Adding constraints on the basis of DP can make the obtained DAG more accurate. The accuracy of constraints depends on expert knowledge. When the number of constraints obtained is large and the accuracy is high, the real DAG can be recovered more accurately.

The simulation studies are not at all convincing. I would like to see more cases and comparison with other algorithms. Compute some performance metrics and show that you do better, in higher dimensional settings, e.g. 50, 100, 200 variables.

Response: This paper proposed a double constrained dynamic programming BN structure learning algorithm, mainly solving the problem that the DP algorithm cannot obtain an accurate structure when the sample size is small. In the algorithm simulation and analysis in Section 4, we can see that when the sample size is 20, both the PBDP-EDGE and PBDP-PATH algorithms can obtain a more accurate structure than the DP algorithm. The algorithm proposed in this paper used the prior as a constraint to limit the planning process of dynamic programming and reduced the planning space. Although the algorithm is within a certain range, the algorithm used the constraint to limit the selection of the optimal parent node to ensure that the optimal structure conforms to the prior. The efficiency and accuracy of BN structure learning are improved, but it is still limited by the DP algorithm, that is, although the algorithm proposed in this paper can obtain the global optimal structure, it is not suitable for large networks.

Round 2

Reviewer 2 Report

I had asked for a specific comment which was not addressed.

The simulation studies are not at all convincing. I would like to see more cases and comparison with other algorithms. Compute some performance metrics and show that you do better, in higher dimensional settings, e.g. 50, 100, 200 variables.